# SARS-CoV-2 Coronavirus Disease in Patients with Underlying Congenital Inferior Vena Cava Anomalies

**DOI:** 10.3390/biomedicines13061336

**Published:** 2025-05-29

**Authors:** Fadia Salman, Pierfrancesco Antonio Annuvolo, Marta Minucci, Francesco Sposato, Ottavia Borghese, Yamume Tshomba

**Affiliations:** 1Unit of Vascular Surgery, Fondazione Policlinico Universitario A. Gemelli I.R.C.C.S., 00168 Rome, Italy; salmanfadia02@gmail.com (F.S.); minucci.marta@gmail.com (M.M.); dott.francescosposato@gmail.com (F.S.); ottaviaborghese@gmail.com (O.B.); yamume.tshomba@policlinicogemelli.it (Y.T.); 2Department of Cardiovascular Sciences, Vascular and Endovascular Surgery School, Faculty of Medicine and Surgery, Università Cattolica del Sacro Cuore, 00168 Rome, Italy

**Keywords:** COVID-19, deep vein thrombosis, anticoagulation, COVID-19-associated coagulopathy, congenital inferior vena cava anomalies

## Abstract

**Background:** COVID-19-related deep vein thrombosis (DVT) in patients with pre-existing congenital anomalies or genetic diseases of the cardiovascular system has been rarely reported, and a clear definition of best treatment in this setting remains undefined. **Methods:** We report the rare case of a 36-year-old male patient affected with a congenital cardiovascular anomaly, presenting extensive venous thrombosis following COVID-19-induced coagulopathy. An insight into current treatment strategies in this setting is also reported. **Results and Conclusions:** COVID-19 disease appears to be a determining factor in the development of extensive DVT in patients with congenital anomalies and genetic disorders. Anticoagulation should be tailored to the individual risk factors, balancing the risk-benefit between prevention of VTE and hemorrhagic complications.

## 1. Introduction

Patients affected with severe or critical COVID-19 may also suffer from cardiovascular complications related to a prothrombotic state [1,2]. Indeed, SARS-CoV-2 acts by binding its spike protein to Angiotensin-Converting Enzyme 2 (ACE2), a membrane-bound aminopeptidase highly expressed in heart and lung cells [3].

This causes endothelial dysfunction aggravated by a dysregulation of the immune system and favors a prothrombotic state with an increased deposition of platelets, circulating inflammatory cells, and protein.

An increased blood viscosity may also lead to venous thromboembolism (VTE) with possible deep vein thrombosis (DVT) and/or pulmonary embolism (PE) [4].

COVID-19-related DVT in patients with pre-existing congenital anomalies or genetic diseases of the cardiovascular system has been rarely reported, and a clear definition of candidates for thrombotic prophylaxis in this setting remains undefined.

Herein, we report the rare case of a 36-year-old male patient affected with a congenital cardiovascular anomaly, presenting extensive venous thrombosis following COVID-19-infection. An insight into current treatment strategies in this setting is also reported.

## 2. Ethical Statement and Patient Consent

All procedures at our institution are performed following the ethical standards set by institutional and national research committees and the 1975 Helsinki Declaration and its amendments. Our institution does not require ethical approval for reporting individual cases or case series. The patient’s consent was obtained for publication of this report and images.

## 3. Case Report

A 39-year-old Caucasian male patient presented to the emergency department of our institution with acute back and abdominal pain, and swelling and edema of the lower limbs with cyanosis. He also reported onset of fever, asthenia, rhinorrhea, and odynophagia two weeks earlier and was tested positive for COVID-19 using a nasopharyngeal PCR test. The SARS-CoV-2 test was repeated at admission but yielded a negative result. He had no fever, and O_2_ saturation was >95% in room air.

Blood tests documented increased D-Dimer (752 ng/mL), C-reactive protein (109 mg/L), Interleukin 6 (18.6 pg/mL) and fibrinogen values (435 mg/dL), Interleukin 1, and TNF α, while other inflammatory and coagulative parameters were within the normal range.

A duplex ultrasound of the lower limbs was performed, and deep bilateral vein thrombosis (DVT) of the common femoral and external iliac veins with possible involvement of the inferior vena cava (IVC) was detected.

The patient denied personal and family history of important thrombotic events. The congenital and acquired thrombophilic risk tests were normal.

A respiratory alkalosis was found at the blood gas analysis so in suspicion of PE, Computed Tomography Angiography (CTA) was performed and did not detect any sign of pulmonary thromboembolism but azygos and hemiazygos vein dilatation (Figure 1) and a congenital abnormality of the IVC, in addition to DVT of iliaco-femoral veins with involvement of the IVC, renal veins, and mesenteric vein (Figure 2). Anticoagulation with low molecular weight heparin (LMWH) 1 mg/kg every 12 h subcutaneously was started.

Symptoms rapidly improved during hospital stay, and the patient was discharged home in good clinical conditions. Oral anticoagulation was continued with rivaroxaban 20 mg daily for 3 months.

At the 3-month follow-up visit, the patient’s clinical status was stable, with persistence of mild edema and swelling of the lower limbs but no pain. Duplex ultrasound showed a partial recanalization of common femoral veins with no improvement at the level of the iliac arteries.

The patient was continued on oral anticoagulant therapy for 12 months, and his clinical condition remained stable.

He is still under an ongoing follow-up.

## 4. Discussion

The coagulopathic profile of COVID-19 has been widely recognized and is largely attributed to a profound dysregulation of the coagulation cascade, primarily driven by an exaggerated inflammatory response known as the “cytokine storm”, with elevated levels of tumor necrosis factor-α (TNF-α), interleukin (IL)-1, IL-6, and other chemokines contributing to endothelial dysfunction, platelet activation, and hypercoagulability [5,6,7,8,9]. This has translated into a high incidence of thrombotic complications among hospitalized patients, including pulmonary embolism (PE) and deep vein thrombosis (DVT), with reported rates of 16.5% and 14.8%, respectively [10].

In the context of viral infections, the thrombogenic potential of SARS-CoV-2 appears particularly pronounced compared to other pathogens such as HIV (1.5% incidence of VTE in non-injecting drug users) [11] or H1N1 influenza (5.9%) [12]. This suggests an intrinsic prothrombotic signature associated with COVID-19, particularly in patients with pre-existing risk factors for venous thromboembolism (VTE), including congenital anomalies of the cardiovascular system (Table 1) [13,14].

Our patient presented with extensive DVT following a recent COVID-19 infection and was subsequently found to have a congenital atresia of the inferior vena cava (IVC). IVC agenesis or atresia is an exceptionally rare condition, with an estimated prevalence of 0.0005–1% in the general population [15]. This anomaly, often underdiagnosed, predisposes to venous stasis and increases the risk of thrombotic events, especially in young patients. While the exact etiology remains uncertain, both congenital malformation and in utero thrombosis have been postulated as causative mechanisms [15,16].

Notably, DVT may occur in patients with IVC anomalies even in the absence of identifiable prothrombotic triggers [17]. However, in the presence of COVID-19-induced coagulopathy, the thrombotic risk may be amplified, suggesting a synergistic interaction between underlying vascular anomalies and systemic inflammatory states.

The recent literature highlights the lack of consensus on optimal prophylactic strategies for VTE in the context of COVID-19. Some studies proposed intensified anticoagulation regimens early in the pandemic, but subsequent trials have emphasized the need for individualized risk stratification based on disease severity, body mass index, renal function, and coagulation markers such as D-dimer [18,19,20,21]. While standard-dose low molecular weight heparin (LMWH) remains the cornerstone for thromboprophylaxis in most hospitalized patients, intermediate or therapeutic dosing should be reserved for selected high-risk populations, such as those with obesity (BMI > 30 kg/m^2^), mechanical ventilation, or markedly elevated D-dimer levels [22,23] (Table 2).

Susen et al. recommend standard LMWH prophylaxis (e.g., enoxaparin 4000 IU daily) for patients with intermediate thrombotic risk, and dose adjustments based on weight or renal insufficiency (e.g., unfractionated heparin for creatinine clearance < 30 mL/min) [23]. Importantly, in the absence of critical illness, therapeutic anticoagulation does not confer added benefit and may increase the risk of bleeding [22,24,25].

Antiplatelet therapy, while explored in the context of long-COVID-related endothelial dysfunction, has not demonstrated efficacy in the acute phase of COVID-19 for VTE prevention [15,17]. Thus, its routine use for prophylaxis in this setting is not supported.

Congenital IVC anomalies are rarely discussed in the COVID-19-related thrombosis literature, but several recent case reports underscore their clinical relevance (Table 3). In one case, a young male developed massive DVT following mRNA vaccination, prompting thrombectomy and lifelong anticoagulation [26]. Other cases were managed conservatively with LMWH followed by oral anticoagulants [27,28].

Given the paucity of robust data, thromboprophylaxis in patients with congenital IVC anomalies should be guided by a thorough evaluation of individual risk factors such as age, immobility, malignancy, and prior thrombotic events. In our case, the patient did not undergo thromboprophylaxis during COVID-19 infection, considering that the anomaly of the vena cava was not yet known and that the COVID-19 infection was not associated with further significant thromboembolic risk factors (age, mildness of the clinical manifestations of the infection, absence of immobility, etc.). The thrombotic event was treated with LMWH in the acute phase, then transitioned to a DOAC despite the atypical localization of the thrombosis, with favorable outcomes.

In summary, congenital IVC anomalies may act as amplifiers of thrombotic risk in the presence of COVID-19-induced coagulopathy. However, the general principles of VTE management in COVID-19 patients—namely, risk stratification, tailored anticoagulation, and cautious duration of therapy—remain applicable. Larger studies are needed to establish specific guidelines for this unique patient subgroup.

**Table 1 biomedicines-13-01336-t001:** Risk factors for venous thromboembolism during COVID-19 infection.

Risk Factors	
Individual risk factors	Old age, reduced mobility, obesity, active cancer, history of previous VTE/PE, and congenital anomalies of cardiovascular system [13,20,24].
COVID Severity	Need for admission in ICU. Need for ECMO [18,19].
Laboratory tests	Elevated D-dimer (>3 μg/mL), C-reactive protein, ferritin, white blood cell count, and IL-6; Lupus Anticoagulant positivity; fibrinogen > 8 g/L [19,20].

VTE, venous thromboembolism; PE, pulmonary embolism; ECMO, extracorporeal membrane oxygenation; ICU, intensive care unit.

**Table 2 biomedicines-13-01336-t002:** Anticoagulation strategies for the prevention of venous thromboembolism in patients with COVID-19 infection.

Clinical Setting	Treatment
Severe COVID-19 (ICU patients)/ventilated patients	Preventive LMWH is recommended [22].
Patients with specific risk factors for VTE	Thromboprophylaxis should be given during the hospital stay and for 1–2 weeks following discharge and continued for at least three months after VTE [20,21,24,25].
Patients affected by chronic kidney disease, AKI, or hepatic injury	Caution should be applied in use of anticoagulation as there is an increased risk of bleeding [18,19,22].

ICU, intensive care unit; VTE, venous thromboembolism; LMWH, low-molecular-weight heparin; AKI, acute kidney injury.

**Table 3 biomedicines-13-01336-t003:** Reported cases of COVID-19-related deep vein thrombosis in patients with congenital inferior vena cava anomalies.

Authors, Year	Case
Jenab et al., 2022 [28]	A 33-year-old man with bilateral DVT and absence of IVC during concurrent COVID-19.
Costanzo et al., 2023 [27]	An 86-year-old woman with recent hospitalization for severe COVID-19 infection. CT of the abdomen and pelvis revealed a duplicated IVC with a thrombus located in the right IVC.
Nam et al., 2022 [26]	Sudden massive DVT in a young man with inferior vena cava anomaly 20 h after the second dose of the mRNA vaccine for COVID-19.

DVT, deep vein thrombosis; IVC, inferior vena cava; CT, computed tomography.

## 5. Results and Conclusions

COVID-19 disease has been shown to induce endothelial cell damage and lead to a hypercoagulable condition even in the absence of pre-existing alterations of the coagulative status, with an increased incidence of DVT or PE in affected patients.

The impact of COVID-19-induced coagulopathy on underlying congenital anomalies and genetic diseases has been rarely reported but appears to be a determining factor in the development of extensive deep vein thrombosis.

There remains great variability in the approach to anticoagulation in COVID-19 which should be customized according to individual risk factors balancing the risk–benefit between the prevention of VTE and hemorrhagic complications.

Further studies are needed to indicate a definitive treatment strategy in this setting.

## Figures and Tables

**Figure 1 biomedicines-13-01336-f001:**
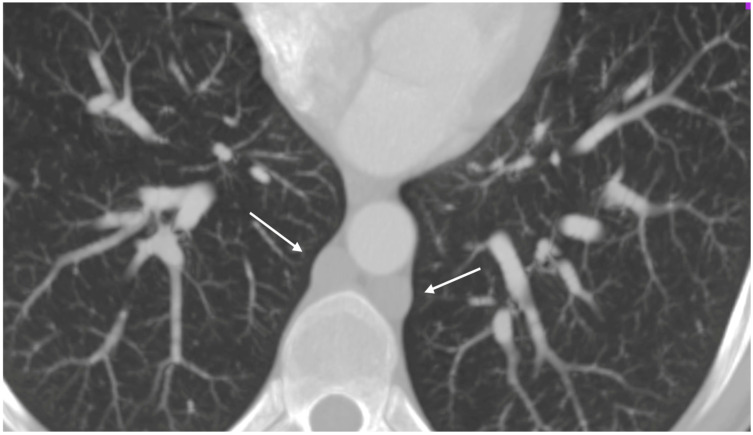
Thoracic angio-TC: dilated azygos and hemiazygos veins (arrows); no pulmonary embolism signs.

**Figure 2 biomedicines-13-01336-f002:**
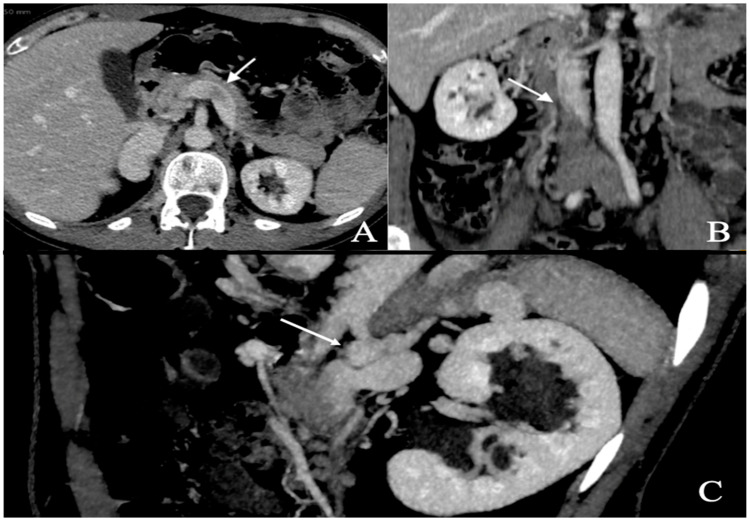
Abdominal angio-TC: extensive deep thrombosis involving femoral and iliac veins, renal veins (**A**—white arrow), IVC (**B**—white arrow), inferior mesenteric vein; abnormal hemiazygos and lumbar vein dilatation (**C**—white arrow).

## Data Availability

The original contributions presented in this study are included in the article. Further inquiries can be directed to the corresponding author.

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
