# Peer review of "SARS-CoV-2 Coronavirus Disease in Patients with Underlying Congenital Inferior Vena Cava Anomalies"

_biomedicines, 2025, doi:10.3390/biomedicines13061336_

Round 1
Reviewer 1 Report (Previous Reviewer 1)
Comments and Suggestions for Authors
Dear Colleagues
The description of the clinical case is definitely more coherent and harmonized even if it still has inaccuracies and some significant mistake.
Overall, the purpose of the work is still unclear, beyond the presentation of the clinical case. In particular, the discussion is not clear whether the intention is to talk about anomalies of the vena cava, prophylaxis of VTE in patients with COVID, or treatment of VTE.
Since the treatment of VTE during SARS-CoV 2 infection is no different from that in uninfected subjects, I believe an analysis in this sense is not useful.
Perhaps more interesting is the discussion on prophylaxis during SARS-CoV 2 infection by relating it to your case: in the first phase of epidemy, different dosages of EBPM have been proposed for different types of patients. What is the validity of these proposals today, after many years of esperience? And would your patient have been indicated for prophylaxis?
The description of different doses of LMWH in relation to weight or renal function are confusing, and: why should there be different indications than in the population without infection?
As far as I know, antiplatelet therapy has been proposed in long-COVID but not in VTE prophylaxis during the acute phase of the infection. (Page 5, lines 15-16).
Overall I think the discussion needs to be reviewed and made more linear
Some detailed suggestions:
Introduction:
The first paragraph (page 1, lines 29-33) seems redundant to me, and the size of the pandemic has no relation to the content of the article
Clinical case:
Page 2, lines 33-35: the thrombosis was probably bilateral, but it is not described
Page 2, line 38: why do you describe the absence of anomalies of factor VII and vWF that would have nothing to do with the thrombosis events?
Page 2, Lines 36-39: it is not necessary to express the individual thrombophilia tests in family members, it is sufficient to say that the congenital and acquired thrombophilic risk tests are normal
Page 2, Line 49: heparin therapy for VTE is 1 mg/Kg x 2/day. What you describe is erroneous, as is the one reported in the discussion (Page 7, line 3)
Page 3, Line 6: you're talking about arteries: why?
Figure 1: the description of the complete computed angiotomography report is not necessary, it is sufficient to indicate the dilatation of azygos and hemiazygos veins
Figure 2: mesenteric arteries are mentioned: why?
Author Response
Comments 1:
Dear Colleagues
The description of the clinical case is definitely more coherent and harmonized even if it still has inaccuracies and some significant mistake.
Overall, the purpose of the work is still unclear, beyond the presentation of the clinical case. In particular, the discussion is not clear whether the intention is to talk about anomalies of the vena cava, prophylaxis of VTE in patients with COVID, or treatment of VTE.
Since the treatment of VTE during SARS-CoV 2 infection is no different from that in uninfected subjects, I believe an analysis in this sense is not useful.
Perhaps more interesting is the discussion on prophylaxis during SARS-CoV 2 infection by relating it to your case: in the first phase of epidemy, different dosages of EBPM have been proposed for different types of patients. What is the validity of these proposals today, after many years of esperience? And would your patient have been indicated for prophylaxis?
The description of different doses of LMWH in relation to weight or renal function are confusing, and: why should there be different indications than in the population without infection?
As far as I know, antiplatelet therapy has been proposed in long-COVID but not in VTE prophylaxis during the acute phase of the infection. (Page 5, lines 15-16).
Overall I think the discussion needs to be reviewed and made more linear
Some detailed suggestions:
Introduction:
The first paragraph (page 1, lines 29-33) seems redundant to me, and the size of the pandemic has no relation to the content of the article
Clinical case:
Page 2, lines 33-35: the thrombosis was probably bilateral, but it is not described
Page 2, line 38: why do you describe the absence of anomalies of factor VII and vWF that would have nothing to do with the thrombosis events?
Page 2, Lines 36-39: it is not necessary to express the individual thrombophilia tests in family members, it is sufficient to say that the congenital and acquired thrombophilic risk tests are normal
Page 2, Line 49: heparin therapy for VTE is 1 mg/Kg x 2/day. What you describe is erroneous, as is the one reported in the discussion (Page 7, line 3)
Page 3, Line 6: you're talking about arteries: why?
Figure 1: the description of the complete computed angiotomography report is not necessary, it is sufficient to indicate the dilatation of azygos and hemiazygos veins
Figure 2: mesenteric arteries are mentioned: why?
Response 1: Dear Reviewer,
We have edited and implemented the discussion as requested.
We have also edited the text within the introduction and the clinical case as suggested.
Reviewer 2 Report (Previous Reviewer 2)
Comments and Suggestions for Authors
In the revised manuscript, the authors revised extensively. I support the acceptance of this manuscript.
Author Response
Comments 2: In the revised manuscript, the authors revised extensively. I support the acceptance of this manuscript.
Response 2: Dear Reviewer, thanks for you work and for your suggestions.
Round 2
Reviewer 1 Report (Previous Reviewer 1)
Comments and Suggestions for Authors
Great work!
The discussion is well conducted with clear logical thinking.
A few more suggestions:
- I would remove the first paragraph completely: Pag 1 lines 29-33
- Page 2 line 13: not COVID 19-induced coagulopathy, but: COVID 19-infection
- Page 2, line 16: materials ant Methods: not applicable, delete it
- Page 2, line 17: 2. Ethical statement...
- Page 2, line 23: enter the title: 3. case report
- Page 2 line 33: ... and deep bilateral vein thrombosis….
- Page 2, lines 40-44: you can eliminate these lines because the concept has already been expressed in lines 39-40 (The congenital and acquired…are normal)
- Page 2, line 49:…IVC, in addition to DVT of iliaco-femoral veins with involvement of IVC, renals veins and mesenteric vein (figure 2)
- Page 4, lines 11-12: may be a “with” is missig after “Cytokine storm”?
- Page 5, line 21-23: after “prior thrombotic events.” : In our case, the patient did not undergo thromboprophylaxis during COVID infection, considering that the anomaly of the vena cava was not yet known and that the COVID infection was not associated with further significant thromboembolic risk factors (age, mildness of the clinical manifestations of the infection, absence of immobility, ect…). The thrombotic event was treated with LMWH in the acute phase, then transitioned to an DOAC despite the atypical localization of the thrombosis, with favorable outcomes
- Figure 1: toracic angio-TC: dilated azygos and hemiazigos veins (arrows), no pulmonary embolism signs
- Figure 2: Abdominal angio-TC: extensive deep thrombosis involving femoral and iliac veins, renal veins (A), IVC (B), inferior mesenteric vein; abnormal hemiazygos and lumbar veins dilatation (C)
Author Response
Comments 1:
Great work!
The discussion is well conducted with clear logical thinking.
A few more suggestions:
- I would remove the first paragraph completely: Pag 1 lines 29-33
- Page 2 line 13: not COVID 19-induced coagulopathy, but: COVID 19-infection
- Page 2, line 16: materials ant Methods: not applicable, delete it
- Page 2, line 17: 2. Ethical statement...
- Page 2, line 23: enter the title: 3. case report
- Page 2 line 33: ... and deep bilateral vein thrombosis….
- Page 2, lines 40-44: you can eliminate these lines because the concept has already been expressed in lines 39-40 (The congenital and acquired…are normal)
- Page 2, line 49:…IVC, in addition to DVT of iliaco-femoral veins with involvement of IVC, renals veins and mesenteric vein (figure 2)
- Page 4, lines 11-12: may be a “with” is missig after “Cytokine storm”?
- Page 5, line 21-23: after “prior thrombotic events.” : In our case, the patient did not undergo thromboprophylaxis during COVID infection, considering that the anomaly of the vena cava was not yet known and that the COVID infection was not associated with further significant thromboembolic risk factors (age, mildness of the clinical manifestations of the infection, absence of immobility, ect…). The thrombotic event was treated with LMWH in the acute phase, then transitioned to an DOAC despite the atypical localization of the thrombosis, with favorable outcomes
- Figure 1: toracic angio-TC: dilated azygos and hemiazigos veins (arrows), no pulmonary embolism signs
- Figure 2: Abdominal angio-TC: extensive deep thrombosis involving femoral and iliac veins, renal veins (A), IVC (B), inferior mesenteric vein; abnormal hemiazygos and lumbar veins dilatation (C)
Response 1: Dear Editor, we have corrected the text as requested. Thanks for your precious work.
This manuscript is a resubmission of an earlier submission. The following is a list of the peer review reports and author responses from that submission.
Round 1
Reviewer 1 Report
Comments and Suggestions for Authors
Dear collegues,
the clinical case would be of interest for thrombosis description in atresia of the inferior vena cava; but overall the presentation is confusing.
It is not clear whether the article is intended to describe the anomalies of the vena cava or the treatment of VTE in a patient with SARS-CoV2 infection.
The article also describes an anomalous diagnostic process and the VTE therapy indicated here does not follow the correct indications. Specifically at least three elements are inadequate:
- Wrong TVP therapy for indication, type and duration (in any case off-label): two months of fondaparinux
- Performing a CT scan not immediately (it would have been logical) but during the FU
- Performing a PET scan: What the reason?
The English is quite fine, just only few improvements
Author Response
Comments 1:
Dear collegues,
the clinical case would be of interest for thrombosis description in atresia of the inferior vena cava; but overall the presentation is confusing.
It is not clear whether the article is intended to describe the anomalies of the vena cava or the treatment of VTE in a patient with SARS-CoV2 infection.
The article also describes an anomalous diagnostic process and the VTE therapy indicated here does not follow the correct indications. Specifically at least three elements are inadequate:
- Wrong TVP therapy for indication, type and duration (in any case off-label): two months of fondaparinux
- Performing a CT scan not immediately (it would have been logical) but during the FU
- Performing a PET scan: What the reason?
Response 1:
Dear Reviewer
The aim of the submitted article is to describe the impact of Sars-CoV2 infection in the development of extensive DVT in patient with underling congenital anomalies like atresia of the inferior vena cava. We have clarified our purpose within the text as suggested.
The patient was discharged home on oral anticoagulant therapy (rivaroxaban 20 mg daily) for 3 months, then continued for 12 months due to persistent symptoms. A total body CTA was performed immediately in the suspect of PE. We have made a correction in the text, with a better description of the diagnosis process, in line with the suggestions.
Reviewer 2 Report
Comments and Suggestions for Authors
In this manuscript, the authors present a case of a COVID patient who developed venous thrombosis post-infection and offer treatment recommendations for similar at-risk patients. The study is valuable, and I support its acceptance with some revisions addressed:
- The manuscript suggests that the mechanism behind COVID-19 inducing DVT may be linked to a cytokine storm. Could the authors clarify whether the patient exhibited signs of pneumonia or elevated cytokine levels? If not, what alternative mechanisms might explain the onset of DVT in patients with congenital anomalies during COVID.
- While discussing the necessity of weighting risk-benefit ratios, the potential risks associated with anticoagulation therapy should be elaborated upon.
- The estimation of the incidence of DVT in COVID patients appears vague. A clearer percentage would aid in understanding the scale of risk. In addition, it would be beneficial if the authors could compare this risk with that associated with other viral infections, such as influenza, to contextualize the findings further.
Author Response
Comments 2:
In this manuscript, the authors present a case of a COVID patient who developed venous thrombosis post-infection and offer treatment recommendations for similar at-risk patients. The study is valuable, and I support its acceptance with some revisions addressed:
- The manuscript suggests that the mechanism behind COVID-19 inducing DVT may be linked to a cytokine storm. Could the authors clarify whether the patient exhibited signs of pneumonia or elevated cytokine levels? If not, what alternative mechanisms might explain the onset of DVT in patients with congenital anomalies during COVID.
- While discussing the necessity of weighting risk-benefit ratios, the potential risks associated with anticoagulation therapy should be elaborated upon.
- The estimation of the incidence of DVT in COVID patients appears vague. A clearer percentage would aid in understanding the scale of risk. In addition, it would be beneficial if the authors could compare this risk with that associated with other viral infections, such as influenza, to contextualize the findings further.
Response 2:
- The patient reported onset of fever, asthenia, rhinorrhea, and odynophagia two weeks earlier, and at the admission blood tests documented increased values of inflammatory parameters. We clarified within the text.
-
We described in the discussion that caution should hence be applied in patients with conditions such as chronic kidney disease, acute kidney injury (AKI) or hepatic injury that present an increased risk for developing hemorrhagic complications (haematomas, haematuria, internal bleeding).
- The incidence of DVT in COVID patients is 14.8% in the meta-analysis cited in the text, which analysed 27 studies involving 3342 patients with COVID-19. We compared the risk of developing DVT in Covid patients to that associated with other infectious diseases. We modified the text as suggested.